# Proteomic Analysis of Differentially Expressed Proteins in A549 Cells Infected with H9N2 Avian Influenza Virus

**DOI:** 10.3390/ijms26020657

**Published:** 2025-01-14

**Authors:** Conghui Zhao, Xiaoxuan Zhang, Huanhuan Wang, Haoxi Qiang, Sha Liu, Chunping Zhang, Jiacheng Huang, Yang Wang, Peilin Li, Xinhui Chen, Ziyi Zhang, Shujie Ma

**Affiliations:** Fujian Province Joint Laboratory of Animal Pathogen Prevention and Control of the “Belt and Road”, College of Animal Sciences, Fujian Agriculture and Forestry University, Fuzhou 350002, China; zhaoconghui19@163.com (C.Z.); zhangxiaoxuan5426@163.com (X.Z.); wanghuan12282487@163.com (H.W.); siriusxx_010106@163.com (H.Q.); m15037362321@163.com (S.L.); m18806027576@163.com (C.Z.); m13395087570@163.com (J.H.); wangyangvv2023@163.com (Y.W.); lpl0051129@163.com (P.L.); 18698331895@163.com (X.C.); 17350500508@163.com (Z.Z.)

**Keywords:** avian influenza virus, H9N2, proteomic analysis, mass spectrometry, bioinformatics analysis, SNAPIN

## Abstract

Influenza A viruses (IAVs) are highly contagious pathogens that cause zoonotic disease with limited availability of antiviral therapies, presenting ongoing challenges to both public health and the livestock industry. Unveiling host proteins that are crucial to the IAV life cycle can help clarify mechanisms of viral replication and identify potential targets for developing alternative host-directed therapies. Using a four-dimensional (4D), label-free methodology coupled with bioinformatics analysis, we analyzed the expression patterns of cellular proteins that changed following H9N2 virus infection. Compared to the control group, the H9N2 infected group displayed 732 differentially expressed proteins (DEPs), with 298 proteins showing upregulation and 434 proteins showing downregulation. Gene Ontology (GO) functional analysis showed that DEPs were catalog in 11 biological processes, three cellular components, and eight molecular functions. Kyoto Encyclopedia of Genes and Genomes (KEGG) pathway analysis revealed that DEPs were involved in processes including cytokine signaling pathways induced by virus infection and protein digestion and absorption. Proteins including TP53, DDX58, and STAT3 were among the top hub proteins in the protein–protein interaction (PPI) analysis, suggesting that these signaling cascades could be essential for the propagation of IAVs. Furthermore, the host protein SNAPIN was chosen to ascertain the accuracy of expression changes identified through a proteomic analysis. The results indicated that SNAPIN was downregulated following infection with IAVs both in vitro and in vivo, which is consistent with the proteomics results, suggesting that SNAPIN may serve as a key regulatory factor in the viral life cycle of IAVs. Our research delineates an extensive interaction map of IAV infection within the A549 cells, facilitating the discovery of pivotal proteins that contribute to the virus’s propagation, potentially offering target candidates to screen for antiviral therapeutics.

## 1. Introduction

Influenza A virus (IAV) is a prevalent zoonotic pathogen, infecting a diverse range of hosts from birds to mammals [1]. A total of 16 distinct hemagglutinin (HA) subtypes and 9 neuraminidase (NA) subtypes of influenza viruses have been detected in avian species, with H17N10 and H18N11 subtypes being exclusively identified in bats [2,3]. Furthermore, the H19 subtype virus has been identified in fecal samples from wild birds based on nucleotide sequencing [4,5]. Despite global efforts to prevent and mitigate the threats posed by IAVs, seasonal influenza epidemics and sporadic pandemics pose substantial challenges to public health. The recent emergence of genetically reassorted viruses, such as H3N8, H5N1, H5N6, H5N8, and H7N9, has caused significant disruption in the livestock sector and poses potential threats to public health [6,7,8,9,10,11,12,13,14]. Therefore, it is crucial that we deepen our understanding of viral pathogenesis and host defense mechanisms to develop more effective countermeasures.

The genome of IAV consists of eight negative-segmented RNAs, encoding the basic polymerase 2 (PB2), basic polymerase 1 (PB1), acidic polymerase (PA), hemagglutinin (HA), nucleoprotein (NP), neuraminidase (NA), matrix protein 1 (M1), matrix protein 2 (M2), nonstructural protein 1 (NS1), and nonstructural protein 2 (NS2). Furthermore, accessory proteins including PB2-S1 [15], PB1-F2 [16], PB1-N40 [17,18], PA-N155 [19], PA-N182 [19], PA-X [20], M42 [21], and NS3 [22] are crucial for viral replication and pathogenesis. IAVs must depend on the host’s cellular components and machineries to facilitate their own propagation [23]. Conversely, the host generates restriction factors, including components of the innate immune system, to suppress the replication of IAVs and mitigate their virulence. The dynamic balance between IAVs and their hosts is a complex network of interactions involving multiple levels of biomolecular mechanisms and evolutionary processes.

H9N2 viruses are prevalent worldwide and have been identified from a variety of wild and domestic avian species [24,25]. As of 10 November 2024, WHO has documented 102 incidents of H9N2 infections in humans, indicating the zoonotic nature of H9N2 IAVs. Additionally, H9N2 viruses have been identified as gene donors that provide internal gene segments to newly emerged viruses, including H3N8, H5N6, H7N9, H10N3, and H10N8, potentially enhancing the risk of pandemics [7,8,12,13,14,26,27]. IAVs can easily acquire the mutations of E627K or D701N in the PB2 protein, which increases the virulence and transmissibility of IAVs in mammals [11,24]. Host factor ANP32A [28,29,30] and virus-encoded protein PA [31] are both instrumental in facilitating the adaptation of avian influenza viruses to mammals. Host restriction factors such as PLSCR1 [32], TRIM25 [33], TRIM35 [34], PIAS1 [35], PKP2 [36], and TUFM [37] are involved in the regulation of IAVs.

Proteomic analysis offers a robust platform for comprehensively identifying differentially expressed proteins (DEPs) following infection of IAVs. These DEPs are frequently engaged in virus–host protein–protein interaction (PPIs) and modulate host signaling pathways, thereby identifying critical host factors for viral propagation. Previous investigations have discovered a limited set of DEPs using two-dimensional electrophoresis (2-DE) combined with mass spectrometry techniques in A549 cells infected with H9N2 viruses [38,39]. To more effectively identify host proteins potentially involved in viral replication, we employed proteomic techniques to analyze the expression patterns of host proteins following infection with the H9N2 virus. In this study, we obtained a proteomic profile in A549 cells following infection with a mammalian-adapted H9N2 virus via a four-dimensional (4D) label-free quantitative strategy coupled with bioinformatics analysis. Our findings may aid in investigating key cellular proteins which modulate the H9N2 virus’s life cycle, as well as in the search for potential drug targets for future research.

## 2. Results

### 2.1. Quantification of Proteins

To ensure the efficiency of viral infection before being subjected to mass spectrometry analysis, viral NP proteins were visualized using an indirect immunofluorescence assay (IFA) in A549 cells infected with the CK/C17-PB2/627K virus at a multiplicity of infection (MOI) of one at 24 h post-inoculation (hpi). NP was detected in nearly all A549 cells infected with the CK/C17-PB2/627K virus after multiple rounds of viral replication, while no NP was detected in the control group inoculated with pure chicken embryo allantoic fluid (Figure 1A). A total of 695,142 spectrograms were acquired using mass spectrometry (MS). Out of these, 332,679 effective spectrograms (48.0%) were utilized for subsequent analyses. The spectrogram analysis revealed that a total of 61,590 peptides were identified, with 59,746 being classified as specific peptide segments. Additionally, 6205 proteins were identified, with 5674 of these being quantifiable (Figure 1B). A principal component analysis (PCA) of all quantifiable proteins showed distinct clusterings between the IAV-infected group and the control group, with a higher protein aggregation degree observed in the CK/C17-PB2/627K virus-infected group (Figure 1C). The Pearson’s correlation coefficient (PCC) quantifies the linear correlation between the two data sets (control and CK/C17-PB2/627K virus infection groups). Scores approaching negative one denote a stronger negative correlation, while scores approaching one denote a stronger positive correlation, and values near zero suggest a lack of correlation. The PCC analysis demonstrated that the quantitative values approached one, indicating a positive correlation between the control group and the CK/C17-PB2/627K virus-infected group (Figure 1D). We plotted the boxplot based on the relative standard deviation (RSD) of protein quantification values among replicate samples. A lower overall RSD indicates superior quantitative consistency. The RSD distribution across replicates underscored the accuracy and reliability of our proteomic data, with the CK/C17-PB2/627K strain-exposed group exhibiting lower RSD values than the negative control group (Figure 1E). These results suggest that the proteomic data in this study are qualified for further analysis.

### 2.2. Quality Control of Data

Quality control analysis confirmed that most peptides identified via mass spectrometry were within the optimal range of 7–20 amino acids. The length distribution of identified peptides complied with the quality control standards, which is crucial for ensuring the accuracy of peptide identification. (Figure 2A). Correspondingly, most proteins were matched with more than two peptides. In the quantitative analysis, a protein associated with multiple distinct peptide segments (or multiple spectra) enhanced the precision and reliability of the quantification results (Figure 2B). According to the shotgun (also known as bottom-up) strategy, the mass spectrometer prioritized scanning peptides with higher concentrations. Therefore, the coverage of a protein was positively associated with the protein’s abundance in the samples. About 21% of these identified proteins had a coverage exceeding 40% (Figure 2C). The identified proteins exhibited a consistent molecular weight distribution across various stages (Figure 2D). These uniform distributions in peptide length, number, coverage, and molecular weight across the identified proteins signify that our results are of high quality and suitable for further detailed analysis.

### 2.3. Analysis of DEPs

Compared to the control group, the CK/C17-PB2/627K group showed 732 DEPs, comprising 298 proteins that were significantly increased (fold change > 1.50) and 434 proteins which were significantly decreased (fold change < 0.67), both at P < 0.05 (Figure 3A). The volcano plot displayed the identified DEPs between the CK/C17-PB2/627K group and the control group (Figure 3B). The heatmap demonstrated the expression patterns of the DEPs (Figure 3C). The top 10 regulated DEPs were shown in Appendix A. Among these proteins, previous studies have shown that CNBP [40] and NRG1 [41] are involved in the regulation of IAVs. Additionally, OSAL and STAT3 are critical factors involved in the antiviral innate immunity induced by IAVs [42]. These results demonstrate that our proteomics data are reliable for further analysis.

### 2.4. Enrichment Analyses of DEPs

To determine if DEPs were significantly enriched in specific functional categories, we performed a functional enrichment analysis of the DEPs using the GO databases. The identified proteins were categorized according to their cellular components based on GO analysis. The GO functional classification revealed that DEPs were cataloged in 22 GO terms, including 11 biological processes, three cellular components, and eight molecular functions. Among the biological process, “cellular process” contained 609 proteins, followed by “biological regulation process” (519 proteins), “metabolic process” (404 proteins), “response to stimulus process” (366 proteins), “multicellular organismal process” (284 proteins), “developmental process” (273 proteins), “localization” (263 proteins), etc. In the cellular process term, the most abundant two terms were “cell” and “intracellular”, with 672 and 637 proteins, respectively. Moreover, “binding” (478 proteins) and “catalytic activity” (264 proteins) were the two top terms in the molecular function group (Figure 4A). Additionally, the functional classification of DEPs was predicted using Clusters of Orthologous Groups of proteins/Eukaryotic Orthologous Groups (COG/KOG) analysis. In the cellular processes and signaling category, the top enrichment was “signaling transduction mechanisms” (89 proteins), while in information storage and processing category, 70 proteins were enriched in the “transcription” function. In the metabolism category, the top three categories were “lipid transport and metabolism”, “carbohydrate transport and metabolism”, and “inorganic ion transport and metabolism” (Figure 4B). These results indicate that DEPs within A549 cells following virus infection could potentially have distinct functions in viral life cycles.

### 2.5. Functional Categories of Proteins

Functional enrichment analyses, including GO annotation and KEGG pathway, were performed to investigate the roles of DEPs in the virus life cycle. In the bubble diagrams, the horizontal axis represents the enrichment factor, while the vertical axis denotes the functional categories or pathways. The enrichment factor was measured via the log_2_-transformed fold change in the proportion of DEPs in that functional type compared to the proportion of identified proteins. The top 20 terms of GO functional enrichment were selected for further analysis based on the enrichment factors (Figure 5A–C). In the enrichment of biological processes, “cytokine-mediated signaling pathways” and “cellular responses to cytokine stimuli” were identified, potentially related to influenza infection. Notably, “synapse assembly” was significantly enriched (Figure 5A). In the cellular component enrichment analysis, “laminin-10 complex” and “CCR4-NOT complex” were the most significantly enriched. Additionally, “lysosomal lumen” and “vacuolar lumen” exhibited relatively higher levels of enrichment and were also identified (Figure 5B). In the molecular function analysis, “glycosphingolipid binding” emerged as the most enriched process, while “semaphorin receptor activity” and “cytokine binding” exhibited notably higher levels of enrichment (Figure 5C). A KEGG pathway analysis indicated that several pathways associated with cell apoptosis and protein degradation were enriched, including “hsa04115 p53 signaling pathway” and “Hsa04974 protein digestion and absorption”, respectively (Figure 5D).

### 2.6. Protein–Protein Interaction (PPI) Network

PPI network analysis was employed to elucidate the functional relationships among DEPs. The diagram depicts a network analysis of DEPs, where each node represents a protein with distinct levels of expression. We submitted 732 DEPs to STRING to visualize their relationships as a network. To enhance the network’s clarity, a threshold of a minimum interaction score of 0.7 was established, and disconnected nodes were omitted. A total of 946 edges comprising the entire network were generated (Figure 6A). The cytoHubba tool was then used to identify the top 40 hub proteins within this network. Within this group of 40 hub proteins, TP53, CDH1, DDX58, BYSL, APP, and STAT3 were identified as central proteins (Figure 6B), potentially playing significant roles in the interactions between the host and IAVs.

### 2.7. SNAPIN Was Downregulated After Infection with IAVs

To verify the accuracy of DEPs identified through proteomic analysis, the downregulated protein SNAPIN was selected for further investigation based on our proteomic data. Autophagy plays a crucial role in the life cycle of IAVs and is regulated by various host factors. In our preliminary literature review, we found that the SNAPIN protein could regulate autophagy [43], but this protein’s role in virus-induced autophagy and its mechanism of action remain unknown. The proteomic results revealed that the SNAPIN protein significantly reduced to 27% compared with the control group at 24 hpi (Figure 7A). To further confirm whether the downregulation of SNAPIN is a ubiquitous response to viral infection, A549 cells were infected with various strains of influenza virus including CK/C17-PB2/627K, WSN, and CK/C88. The results showed that the mRNA levels of SNAPIN were decreased at 24 hpi following infection with CK/C17-PB2/627K (Figure 7B), WSN (Figure 7C), and CK/C88 (Figure 7D). The quantitative PCR (qPCR) results demonstrated a marked reduction in the mRNA levels of SNAPIN 24 hpi following infection with the CK/C17-PB2/627K, WSN, and CK/C88 viruses (Figure 7E–G). These results imply that the decrease in SNAPIN protein levels following IAV infection may be attributable to the downregulation of its mRNA, which is consistent with the proteomic data. To evaluate the in vivo expression alterations of SNAPIN following viral infection, three mice were inoculated with the WSN virus at a dose of 10^6^ EID_50_/50 μL. At three days post-inoculation (dpi), lungs were collected for the analysis of SNAPIN expression levels using qPCR and Western blot. The findings demonstrated a significant downregulation of SNAPIN expression at the mRNA (Figure 7H) and protein levels (Figure 7I) at 3 dpi, with β-actin as the loading control. These results indicate that SNAPIN was downregulated following infection with IAVs both in vitro and in vivo, suggesting that SNAPIN could potentially be a crucial factor in the viral life cycle of IAVs.

## 3. Discussion

IAVs pose significant threats to public health and the livestock industry due to their high contagiousness and the limited availability of effective antiviral therapies. To further decipher host components implicated in viral life cycles, we utilized a 4D label-free proteomics approach coupled with bioinformatics analysis to identify potential host proteins that are involved in the regulation of IAV life cycles. The spectrogram analysis revealed 5674 quantifiable proteins and 732 DEPs that were altered after infection with the CK/C17-PB2/627K avian influenza virus. These DEPs are frequently enriched in specific functional categories and engage in virus–host PPIs to modulate host signaling pathways. Our study may provide insights into viral replication mechanisms and potential targets for host-directed therapies.

Proteomic technology has played critical and essential roles in revealing the interplay between IAVs and host proteins. Haas et al. conducted a proteomic analysis to screen for broad-spectrum antiviral proteins, and sixteen compounds were shown to inhibit IAVs replication by targeting host proteins, two of which targeting CDK2 and FLT3 display antiviral activity across IAVs and coronaviruses [44]. Another proteomic analysis of lung tissues from chicken infected with the highly pathogenic avian influenza (HPAI) H5N1 virus at different time points postinfection revealed the activation of TLR, RLR, NLR, and JAK-STAT signaling pathways. These signaling pathways are related to the cytokine storm effects observed in chickens infected with the HPAI H5N1 virus [45]. In addition, proteomic analysis was employed to identify DEPs following infection with H5 and H9 IAVs in cellular organelles like mitochondrion [39]. These studies suggest that viral infection may optimize its living environment and replication conditions by affecting the expression of cellular proteins. Some proteins, such as mGluR2 [46], ABTB1 [47], and FFAR2 [48], are activated to promote virus replication in cells following IAV infections. Conversely, a series of cellular defense mechanisms, including the innate immune systems and some host proteins, is triggered by activating some signaling pathways to restrict virus replication [32,34,35,42,49,50].

A variety of host factors, including mRNAs, proteins, and non-coding RNAs, contribute to the regulation of IAVs, typically exhibiting altered expression patterns upon viral infections [51,52,53]. Identifying DEPs in IAV-infected cells is a widely employed approach for screening potential regulatory protein candidates. Despite the rapid acquisition of numerous proteins with altered expression through proteomics, challenges remain in identifying effective proteins that influence viral replication. To overcome this obstacle, we developed a screening process for potential DEPs involved in the life cycle of IAVs. Firstly, we assess the expression levels of the DEPs using qPCR and Western blotting techniques to verify the consistency with proteomic data. Secondly, DEPs are overexpressed or knocked down via transfection with specific eukaryotic expression plasmids or siRNAs, respectively. Subsequently, these cells are infected with a HiBit-tagged reporter IAV, and luminescence is determined using a microplate luminometer. The luminescence intensity is directly proportional to the viral replicative ability. Thus, it is anticipated that we may identify crucial proteins that impact viral replication.

Our proteomics analysis has provided a comprehensive overview of the protein expression alterations triggered by viral infections. The DEPs identified in this study reveal valuable perspectives in the dynamics of host–virus interaction. Our GO functional analysis demonstrated that these DEPs were involved in a wide array of biological processes, cellular components, and molecular functions. The “cytokine-mediated signaling pathways” and “cellular responses to cytokine stimuli” in the GO analysis were enriched, and “cytokine binding” showed notably higher levels of enrichment in the KEGG analysis, indicating that the immune system was stimulated upon IAV infection, which is consistent with previous studies [38,53]. Additionally, several pathways related to protein degradation were enriched, including the “ hsa04115 p53 signaling pathway” and “ Hsa04974 protein digestion and absorption”, in the KEGG pathway analysis. These findings suggest that IAV may hijack the host’s protein degradation machinery to facilitate viral replication. A previous screen has shown that TBC1D5 can restrict the replication of IAVs. Conversely, the viral M2 protein can disrupt the binding between TBC1D5 and Rab7, thereby reducing Rab7 activation and arresting lysosomal fusion, which promote M2 to the plasma membrane to enhance virus budding and growth [54]. Hub proteins in proteomics are defined as proteins that have a high degree of connectivity within PPI networks. These proteins engage in a significant number of interactions and are essential in the organization and the function of cellular networks. TP53, an important transcription factor that regulates a range of biological processes including innate immunity and cell apoptosis, is the most highly ranked among the top 40 hub proteins of the PPI network. Previous investigations have indicated that TP53 is involved in the regulation of cell death induced by the influenza virus and the modulation of the interferon response [55,56]. Proteins such as DDX58 and STAT3, which are among the top hub proteins, suggest that innate immunity signaling pathways are crucial for the replication of IAVs. Furthermore, proteins like SQSTM1/p62 and ATG5 are instrumental in the regulation of autophagy [57,58].

In our proteomic data, we found that SNAPIN was significantly downregulated during IAV infection. As a significantly down-regulated protein identified through proteomic analysis (with its expression level dropping to 27% of that in the control group), SNAPIN has been reported to interact with SNAP25, contributing to the formation of SNARE complex, which subsequently regulates the release of synaptic vesicles [59,60]. Studies in recent years have shown that SNAPIN can interact with viral proteins during viral infection, affecting virus replication and spread. The GP5/M proteins of the porcine reproductive and respiratory disorder syndrome virus (PRRSV) employ SNAPIN for intracellular transport and membrane fusion [61]. The interaction between SNAPIN and Human cytomegalovirus (HCMV) has also attracted attention. Studies have shown that multiple proteins encoded by HCMV (including PUL130, UL142, and UL70) can bind to SNAPIN, thereby affecting virus replication [62,63,64]. Furthermore, SNAPIN is crucial in modulating the process of cell autophagy [43,65,66]. Our investigations revealed that the expression of SNAPIN was notably diminished following infection with various strains of IAVs both in vivo and in vitro. The consistent downregulation of SNAPIN across different IAV strains in cells and in mouse models indicates a conserved role for SNAPIN in the viral life cycle. The significant reduction in SNAPIN at both the mRNA and protein levels postinfection may suggest its potential as a biomarker for the detection of IAVs. It is imperative to elucidate the molecular mechanisms underlying the roles of SNAPIN in regulating IAVs in further investigations. Our proteomic network offers a valuable resource for understanding the complex molecular interactions between the virus and host.

## 4. Materials and Methods

### 4.1. Ethics Statement and Facility

The experiments involving live IAVs were conducted in the enhanced biosafety level 2 (BSL2+) facility in Fujian Agriculture and Forestry University. The protocol for mouse studies was approved by the Committee on the Ethics of Animal Experiments of Fujian Agriculture and Forestry University (PZCASFAFU21001).

### 4.2. Cells and Virus

Human lung adenocarcinoma epithelial A549 cells (CCL-185, ATCC, Manassas, VA, USA) were cultured in Dulbecco’s modified Eagle’s medium (DMEM) (11965092, Thermo Fisher Scientific, Waltham, MA, USA) supplemented with 10% fetal bovine serum and antibiotics in a humidified incubator with 5% CO_2_. The H9N2 (A/Chicken/Fujian/C17/2020) virus, a cell-adapted strain with a PB2 E627K mutation, is designated as CK/C17-PB2/627K. The H1N1 virus (A/WSN/1933) is abbreviated as WSN, and the H3N8 virus (A/Chicken/Fuzhou/C88/2021) is abbreviated as CK/C88. The viral titers of CK/C17-PB2/627K, WSN, and CK/C88 were 1.7 × 10^8^ PFU/mL, 2.4 × 10^8^ PFU/mL, and 2.6 × 10^7^ PFU/mL, respectively. Virus stocks were propagated in 10-day-old specific pathogen-free (SPF) embryonated chicken eggs and stored at −80 °C.

### 4.3. Cell Infection

A549 cells were grown in 10 cm dishes and inoculated with the CK/C17-PB2/627K virus at an MOI of one for 1 h, followed by three washes with PBS. For proteomic analysis, the control group was treated with an equal dilution of pure chicken embryo allantoic fluid, compared to the CK/C17-PB2/627K virus stock. The cells were then supplemented with DMEM containing 0.125 mg/mL tosylsulfonyl phenylalanyl chloromethyl ketone (TPCK)-treated trypsin and incubated at 37 °C. Cells were harvested 24 hpi for mass spectrometry analysis. All samples for proteomic analysis were prepared with three replicates to ensure that the obtained data could be subjected to biological statistical analysis. A549 cells were infected with the indicated viruses at an MOI of one for the detection of SNAPIN using nucleic acid electrophoresis, qPCR, and Western blot.

### 4.4. Extraction and Digestion of Proteins

The samples were gently warmed to room temperature. Then, four times the volume of lysis buffer (containing 8 M urea and 1% protease inhibitor cocktail) was added to the samples, followed by ultrasonic lysis (Ningbo Scientz Biotechnology Co., Ltd., Ningbo, China). The remaining cellular debris was removed via centrifugation at 12,000× *g* for 10 min at 4 °C. The supernatant was transferred to a fresh centrifuge tube, and the protein concentration was determined using a BCA assay kit according to the manufacturer’s protocols. Equal amounts of proteins from each sample were subjected to enzymatic digestion, and the volume was adjusted to match the lysis buffer. Trichloroacetic acid (TCA) was added to the samples at a final concentration of 20%, vortexed to mix evenly, and precipitated at 4 °C for 2 h. Samples were centrifuged at 4500× *g* for 5 min. The supernatant was carefully removed, and the precipitate was washed three times with prechilled acetone. Once the precipitate was dried, Tetraethylammonium bromide (TEAB) was added to achieve a final concentration of 200 mM. The precipitate was then dispersed using ultrasonication, and trypsin was added at a ratio of 1:50 (protease to protein, mass to mass) for incubation overnight. Subsequently, dithiothreitol (DTT) was added to the samples to achieve a final concentration of 5 mM. A reduction reaction was performed at 56 °C for 30 min. Following this, iodoacetamide (IAA) was added to the samples for a final concentration of 11 mM, and the mixture was incubated in the dark at room temperature for 15 min.

### 4.5. Liquid Chromatography-Mass Spectrometry Analysis

The peptide segments were separated using an ultra-high-performance liquid chromatography (UHPLC) system, followed by ionization in an NSI ion source in Jingjie PTM BioLabCo., Ltd., (Hangzhou, China). The separated peptides were analyzed using an Orbitrap Exploris™ 480 mass spectrometer (Thermo Fisher Scientific, Waltham, MA, USA). The ion source was set to a voltage of 2.3 kV, with the FAIMS compensation voltage (CV) adjusted to −45 V and −65 V. High-resolution Orbitrap analysis was conducted to detect and analyze both peptide parent ions and their resulting secondary fragments. The primary mass spectrometry parameters were set with a scanning range of 400 to 1200 *m*/*z* and a resolution of 60,000. In the secondary mass spectrometry analysis, the scan range was initiated at a fixed starting point of 110 *m*/*z*, with a resolution set at 15,000, and TurboTMT was deactivated. Data acquisition was performed using a data-dependent acquisition (DDA) program. Following the first-level scan, the top 25 peptide precursor ions exhibiting the highest signal intensity were sequentially selected and subjected to HCD collision cell for fragmentation at an energy setting of 27. Thereafter, the second-level ions were also analyzed sequentially via mass spectrometry. To optimize the mass spectrometer’s efficiency, the automatic gain control (AGC) was set to 100%. Furthermore, the signal threshold was set at 5 × 10^4^ ions/s; the maximum injection time was set to 50 ms, and the dynamic exclusion duration for the tandem mass spectrometry scan was configured to 20 s to avoid repeated acquisition of precursor ions.

### 4.6. Protein Quantification and Criteria for Protein Identification

The MS/MS data were analyzed utilizing the MaxQuant search engine (v.1.5.2.8). Tandem mass spectra were queried against Homo_sapiens_9606_SP_20201214.fasta database concatenated with the reverse decoy database. Trypsin/P was specified as the cleavage enzyme, allowing up to 4 missing cleavages. The mass tolerance for precursor ions was set as 20 ppm in first search and 5 ppm in main search, with a fragment ion tolerance of 0.02 Da. Carbamidomethyl on cysteine was specified as a fixed modification, while protein N-terminal acetylation, methionine oxidation, and deamidation (NQ) were set as variable modifications. The false discovery rate (FDR) was adjusted to less than 1%, and the minimum score for modified peptides was set to a value above 40. We changed the label-free quantitation (LFQ) intensity (I) level of the proteins in different samples through centralization to obtain the relative quantitative value (R) of the protein. The calculation formula is R*_ij_* = I*_ij_*/Mean (I*_j_*), where *i* represents the sample, and *j* denotes the protein. A549 cells treated with pure chicken embryo allantoic fluid served as the control samples, and those infected with CK/C17-PB2/627K were the treated samples. For repeated experiments, the initial step involves calculating the average relative quantification value for each protein across all repetitions. Subsequently, the ratio of these average relative values between the two sample sets is determined. The computed ratio serves as the definitive measure of differential expression for the protein when comparing the two sample groups. To assess the statistical significance of differential expression, a two-sample, two-tailed *t*-test was conducted to calculate the *p* value. Three standard biological replicates were utilized to ascertain whether the quantitative results demonstrated statistical consistency. Pearson’s correlation coefficient (PCC), principal component analysis (PCA), and relative standard deviation (RSD) were employed to assess repeatability.

### 4.7. Bioinformatics Analysis

Fold change > 1.50 or <0.67 and *p* < 0.05 were set as the significance thresholds for DEPs. Based on the terms defined in the Gene Ontology (GO) database (http://geneontology.org/, accessed on 21 January 2021), DEPs were categorized into three categories: biological process, cellular component, and molecular function. The Kyoto Encyclopedia of Genes and Genomes (KEGG) database (www.genome.jp/, release version 97.0, accessed on 21 January 2021) was utilized to annotate protein pathways. Firstly, the KEGG online service tool KAAS was employed to annotate the proteins in the KEGG database. Subsequently, the annotation results were mapped onto the KEGG pathway database using the KEGG mapper tool. The GO and KEGG enrichment of the DEPs against all proteins of the species database was assessed using a two-tailed Fisher’s exact test. The pathway with a corrected *p* < 0.05 is considered statistically significant.

A total of 732 DEPs were analyzed using the STRING platform (https://cn.string-db.org/, Version: 12.0, accessed on 21 November 2024). Then, the PPI network was constructed with the following parameters: the network type was designated to “full STRING network”; the meaning of network edges was defined to “evidence”; active interaction sources included “Textmining, Experiments, Databases, Co-expression, Neighborhood, Co-occurrence, Gene Fusion”; and the minimum required interaction score threshold was set to 0.7. Disconnected nodes within the network were removed. To identify potential hub proteins, the cytoHubba plug-in within the Cytoscape software (version 3.10.3) was used to compute the nodes’ scores. Based on the degree algorithm, 40 proteins with the highest scores were identified and designated as hub proteins.

### 4.8. Mice Study

Six-week-old female BALB/c mice were purchased from Zolgene Biotechnology Co., Ltd., Fuzhou, China. Six mice were randomly divided into two groups evenly. Three mice were lightly anesthetized with CO_2_ and intranasally inoculated with 10^6^ EID_50_ of the WSN virus or PBS (as a control) in a volume of 50 μL and euthanized at 72 hpi. Lungs were collected for the detection of SNAPIN using qPCR and Western blot.

### 4.9. Reverse Transcription, PCR, and Quantitative PCR (qPCR)

A549 cells were inoculated with the indicated viruses at an MOI of 0.1. At 24 hpi, total RNA was extracted using NucleoZOL (740404, Macherey-Nagel, Germany) according to the manufacturer’s instructions. RT-PCR and qPCR for detecting SNAPIN were performed with the following primers: forward primer 5′-AGGAACGACTGAGACGGCTAA-3′ and reverse primer 5′-GGTAAATTCCCGAATCCAGCATT-3′. RT-PCR for detecting NP was performed with the following primers: forward primer 5′-TCAGCATACAACCTACGTTC-3′ and reverse primer 5′-GCATTGTCTCCGAAGAAATAAG-3′. RT-PCR for detecting GAPDH was performed with the following primers: forward primer 5′- AGGTGAAGGTCGGAGTCAACG-3′ and reverse primer 5′-TGGAAGATGGTGATGGGATTTC-3′. Reverse transcription of SNAPIN mRNA was performed using a mixture of oligo dT and random primers, with a HiScript Q RT SuperMix for qPCR (+gDNA wiper) (R123, Vazyme, Nanjing, China) according to the supplied protocol. The SNAPIN gene was amplified via PCR using 2 × Taq Plus Master Mix II (P213, Vazyme, Nanjing, China) and detected via nucleic acid electrophoresis. qPCRs were performed using the ChamQ Blue Universal SYBR qPCR Master Mix (Q312, Vazyme, Nanjing, China) according to the manufacturer’s instructions. GAPDH was used as an endogenous control for mRNA normalization, and relative expression levels for each gene were calculated using the 2^−∆∆Ct^ method.

### 4.10. Western Blotting

Proteins were separated via SDS-PAGE and then blotted onto PVDF membranes (ISEQ00010, Merck-Millipore, Burlington, MA, USA). The mouse anti-SNAPIN monoclonal antibody (mAb) was purchased from Santa Cruz Biotechnology Inc. (sc-514675, Santa Cruz Biotechnology, Dallas, TX, USA). Our lab-generated rabbit anti-NP pAb was used in our Western blot analysis [67]. After blocking with 5% skim milk in PBS, membranes were incubated for 1 h at room temperature with the diluted primary antibody. After incubation with a secondary antibody, the blots were visualized using an Odyssey CLX infrared imaging system (Li-Cor Biosciences, Lincoln, NE, USA).

### 4.11. Indirect Immunofluorescence Assay

A549 cells were cultured on glass-bottom dishes and infected with CK/C17-PB2/627K at an MOI of one. The control group was treated with an equal dilution of pure chicken embryo allantoic fluid. The cells were fixed with 4% paraformaldehyde in PBS for 15 min at 24 hpi and permeabilized with 0.5% Triton X-100 in PBS for 30 min. After being blocked with 5% bovine serum albumin (BSA) (37525, Thermo Fisher Scientific, Waltham, MA, USA) in PBS, the cells were incubated with rabbit anti-NP pAb (1:100) for 1 h. Following three washes with PBS, the cells were incubated for 1 h with the secondary antibody. The cells were then washed three times with PBS and incubated with 4′,6-diamidino-2 phenylindole (DAPl) (S36964, Thermo Fisher Scientific) for 15 min to stain the nuclei. The cells were visualized with the confocal microscope (Leica Stellaris 8 FALCON, Leica Microsystems, Wetzlar, Germany).

### 4.12. Statistical Analysis

Statistical significance between the experimental group and the control group in our qPCR analysis was determined using the one-tailed unpaired *t* test. **, *p* < 0.01.

## 5. Conclusions

Our proteomic analysis reveals that host proteins participate in regulating IAV replication, with 298 proteins upregulated and 434 downregulated. Functional classification and enrichment analysis of these DEPs indicate that IAVs trigger the innate immune response and engage in protein digestion and absorption processes. However, it is important to acknowledge the limitations of our research. The interaction between IAVs and the host is a dynamic process. A proteomic analysis of samples collected 24 hpi was performed in this study; thus, these interaction networks may not fully represent the dynamic nature of the virus–host interaction. Despite identifying numerous DEPs using proteomic techniques, their involvement in the viral life cycle and their specific roles require additional investigation. Further characterization of host proteins critical for viral replication could reveal potential targets for the development of broad-spectrum antiviral therapies. We propose that modulating the expression or activity of these DEPs could offer a novel avenue for antiviral treatment. Specifically, the downregulation of SNAPIN upon IAV infection suggests a potential role in viral pathogenesis. Furthermore, investigating the cross-reactivity of these proteins with other viral strains could evaluate their potential in broad-spectrum antiviral therapies. Our study presents a detailed network for IAV infection in A549 cells, facilitating the screening of key host proteins which regulate viral replication, potentially offering functional targets for the development of broad-spectrum antiviral therapies.

## Figures and Tables

**Figure 1 ijms-26-00657-f001:**
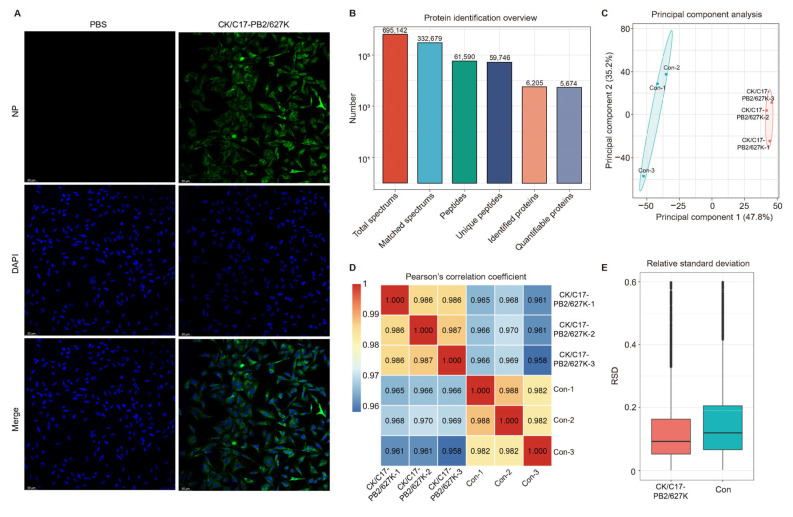
Overview of the mass spectrometry results. (**A**) The infection efficiency was determined using IFA. A549 cells were infected with CK/C17-PB2/627K at an MOI of one and NP protein was visualized with a confocal microscope 24 hpi. Nuclei were stained with DAPI. The white scale bar denotes 50 μm. (**B**) Bar chart illustrating the comparative detection of peptides or proteins in the IAV group and the control group. (**C**) The PCA of protein quantification across all samples. (**D**) Heatmap depicting the PCC values between all sample pairs. This metric assesses the linear relationship between data pairs. Scores approaching negative one reflect a more pronounced inverse relationship; scores approaching one reflect a more pronounced direct relationship, and scores around zero indicate no significant linear association. (**E**) A boxplot of the RSD values of protein quantification among replicate samples. A lower overall RSD signifies superior quantitative reproducibility.

**Figure 2 ijms-26-00657-f002:**
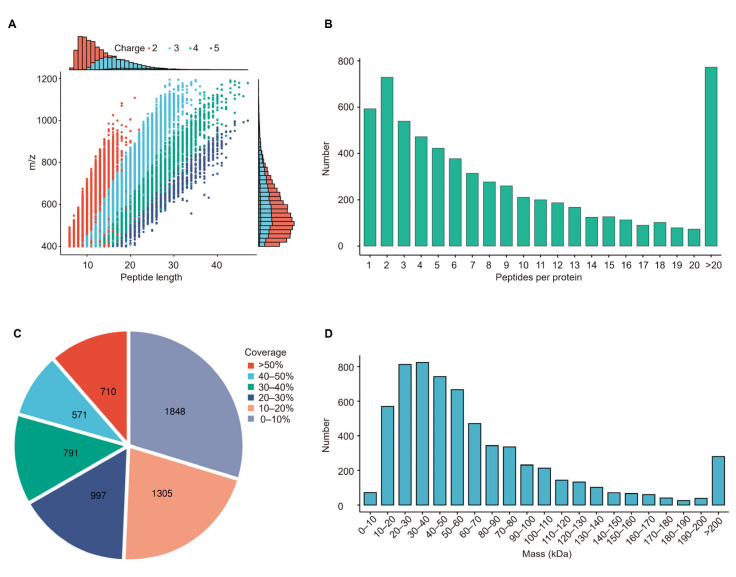
Peptide distribution of proteomic data. (**A**) Peptide length distribution of all identified peptides. (**B**) Peptide number distribution. (**C**) Protein coverage distribution. (**D**) Protein molecular weight distribution.

**Figure 3 ijms-26-00657-f003:**
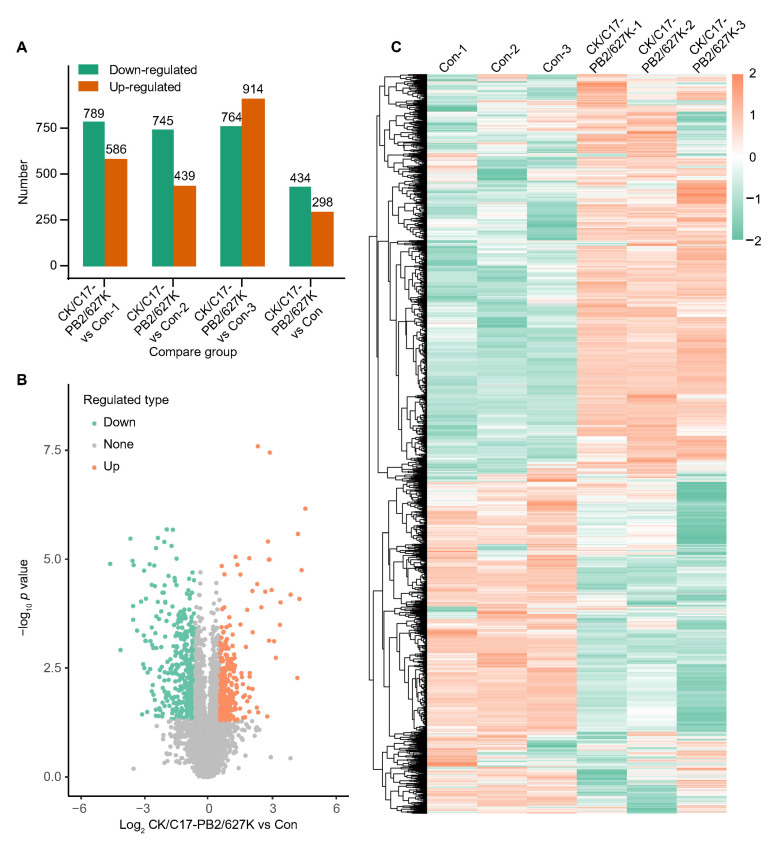
Identification of DEPs between the CK/C17-PB2/627K group and the control roup. (**A**) The total number of upregulated and downregulated DEPs. (**B**) A volcano plot of the identified DEPs between the IAV group and the control group. The upregulated DEPs are indicated by orange dots, downregulated DEPs by cyan dots, and non-varied proteins by gray dots. The *X*-axis corresponds to the fold change in DEPs identified when comparing the CK/C17-PB2/627K group versus the control group, and the *Y*-axis corresponds to transformed *p* values. (**C**) The heatmap shows the hierarchical clustering of samples and DEPs. The high and low expressions of DEPs in different samples are shown with orange and cyan, respectively.

**Figure 4 ijms-26-00657-f004:**
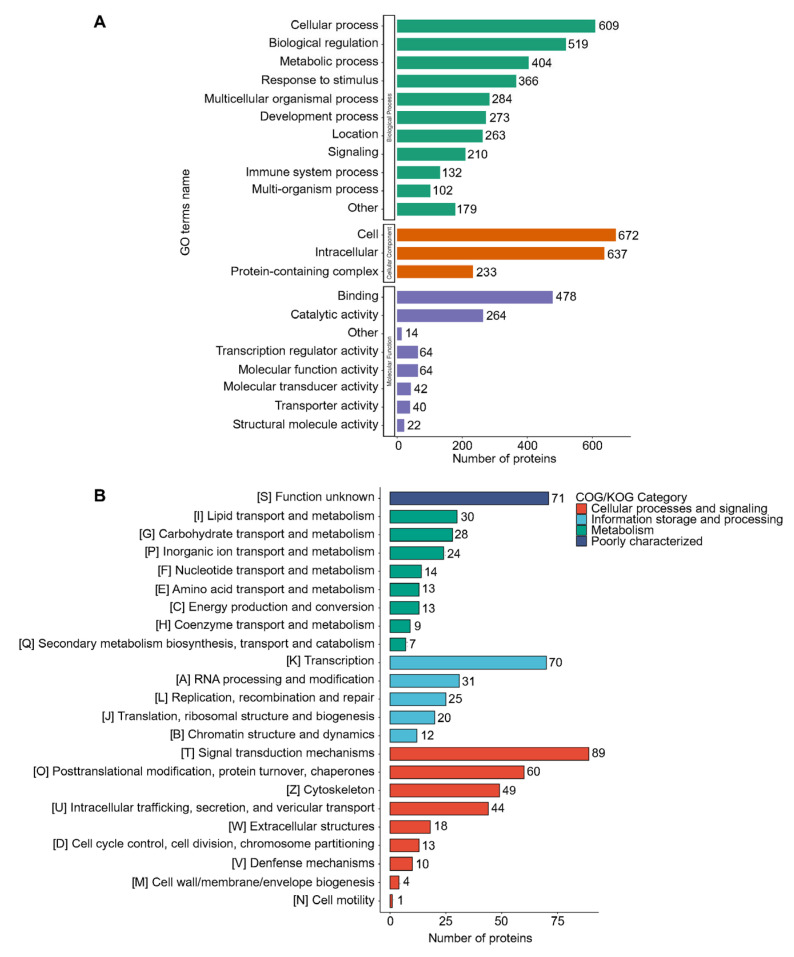
Enrichment analyses of DEPs. (**A**) GO analysis for the identified DEPs. The DEPs were annotated into three categories based on GO terms, including biological processes, cellular components, and molecular functions. (**B**) COG/KOG functional classification analysis of DEPs. The DEPs were aligned against the COG/KOG database and classified into 23 functional clusters. Each bar represents the number of DEPs.

**Figure 5 ijms-26-00657-f005:**
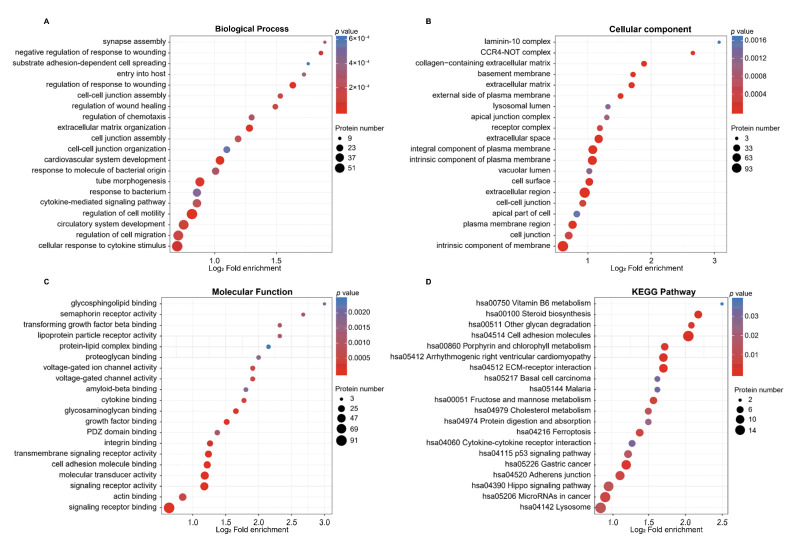
Functional categories of DEPs. Bubble diagrams display biological processes (**A**), cellular components (**B**), molecular functions (**C**), and KEGG pathways (**D**) for significantly enriched DEPs. The color of the circles indicates the enrichment significance *p* value, and the size of the circles represents the number of DEPs in the functional category or pathway.

**Figure 6 ijms-26-00657-f006:**
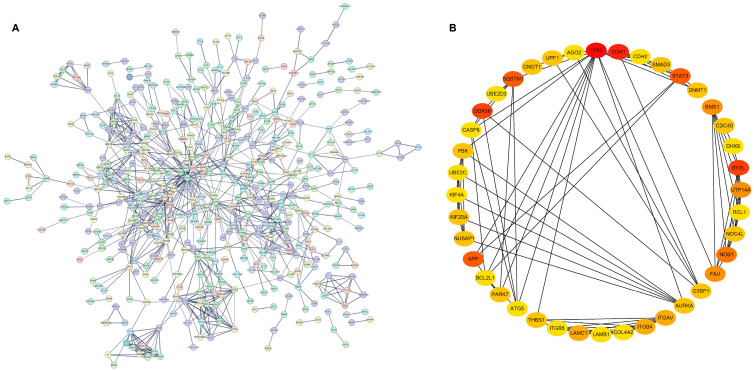
Identification of the hub genes. (**A**) PPI network was performed using the STRING platform. (**B**) Forty hub proteins were identified using cytoHubba. The color intensity of the nodes corresponds to their scores in the degree algorithm; a darker shade signifies a higher score.

**Figure 7 ijms-26-00657-f007:**
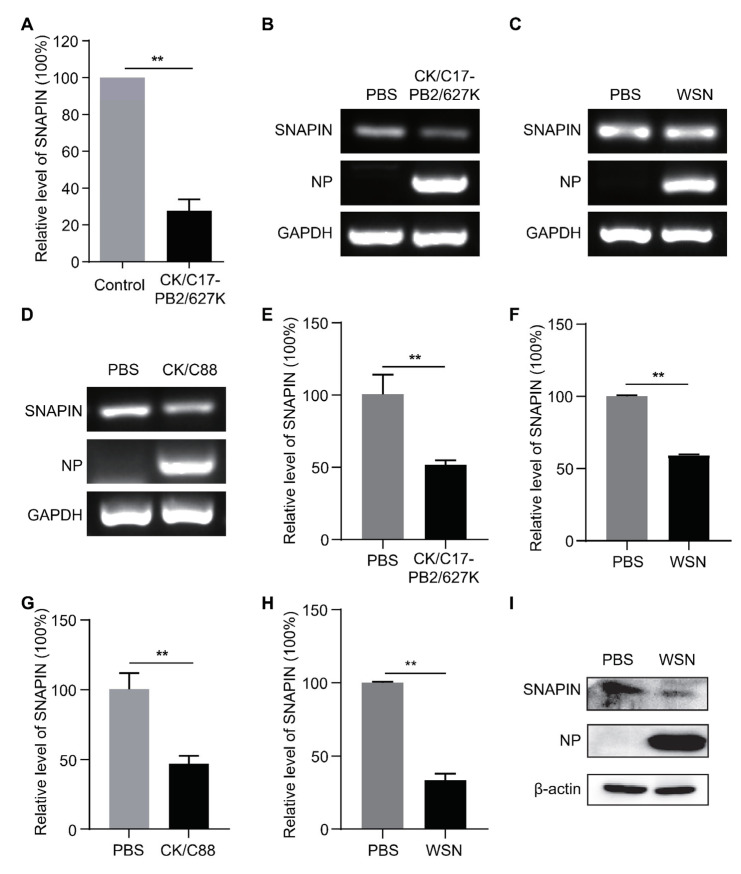
SNAPIN was downregulated following infection with IAVs. (**A**) Downregulation of SNAPIN in A549 cells infected with the CK/C17-PB2/627K virus, determined using proteomic data. Nucleic acid electrophoresis analysis of SNAPIN in A549 cells 24 hpi following infection with the CK/C17-PB2/627K (**B**), WSN (**C**), and CK/C88 (**D**) viruses. qPCR analysis of SNAPIN in A549 cells following infection with the CK/C17-PB2/627K (**E**), WSN (**F**), and CK/C88 (**G**) viruses. (**H**) qPCR and Western blot (**I**) analyses of SNAPIN in mouse lungs following WSN virus infection at 72 hpi. Statistical significance between the experimental group and the control group was determined using the one-tailed unpaired *t* test. **, *p* < 0.01.

## Data Availability

The data supporting the conclusions of this article will be available from the corresponding author upon request.

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
