# Peer review of "Proteomic Analysis of Differentially Expressed Proteins in A549 Cells Infected with H9N2 Avian Influenza Virus"

_ijms, 2025, doi:10.3390/ijms26020657_

Round 1

Reviewer 1 Report

Comments and Suggestions for Authors

Lines 122-126: The presentation of DEP numbers (732 DEPs: 298 increased, 434 decreased) should be accompanied by the statistical criteria used for classification in this initial mention.

Lines 165-180: The functional enrichment analyses results need clearer organization. The "top 20 terms" are mentioned but the selection criteria for determining "top" status isn't explained.

Lines 201-220: The rationale for selecting SNAPIN specifically for validation among the many DEPs identified isn't adequately explained. The authors should justify this choice.

Lines 335-343: The cell infection protocol description lacks important details:

- Specific viral titers used

- Standardization method for MOI across different virus strains

- Controls for viral infection efficiency

Lines 366-383: The mass spectrometry methodology section could be improved by:

- Including details about sample preparation reproducibility

- Explaining the choice of MS parameters

- Describing quality control measures for MS data acquisition

Lines 384-406: The protein quantification section needs:

- Clearer explanation of the normalization process

- More detail about how missing values were handled

- Better description of the statistical approach for determining significance

Lines 407-421: The bioinformatics analysis section requires:

- Version numbers for all software used

- Parameters used for network construction

- Cutoff values for significance in enrichment analyses

Lines 422-426: The mouse study description lacks:

- Power calculation for sample size determination

- Details about randomization

- Specifics about monitoring criteria

Lines 427-437: The PCR methodology needs:

- Primer sequences

- PCR efficiency values

- Details about normalization controls

Lines 438-446: The Western blot section should include:

- Information about loading controls

- Antibody validation details

- Quantification methodology

Lines 450-459: The conclusions section could be strengthened by:

- Addressing study limitations

- Discussing potential therapeutic implications

- Suggesting specific future research directions

Lines 468-471: The funding statement should clarify the role of each funding source in the study design and execution.

Lines 94-102: Figure 1 requires:

- Higher resolution images

- Better explanation of the statistical analysis

- Clearer labeling of axes

Lines 182-185: Figure 5 bubble diagrams need:

- Clearer size legends

- More detailed explanation of the enrichment scores

General formatting issues:

- Inconsistent abbreviation usage throughout the manuscript

- Some figure legends lack sufficient detail

- References need standardization in format

Author Response

For research article

Response to Reviewer 1 Comments

1. Summary

We thank the reviewers for a very careful review and thoughtful comments. These comments are very helpful. The revisions made as per the reviewers’ suggestions have significantly improved the quality of our manuscript. Below are point-by-point responses to the reviewers’ comments. The corresponding revisions in track changes are in the re-submitted files. We hope that the manuscript is now suitable for publication in IJMS.

2. Questions for General Evaluation

Reviewer’s Evaluation

Does the introduction provide sufficient background and include all relevant references?

Yes

Is the research design appropriate?

Yes

Are the methods adequately described?

Can be improved

Are the results clearly presented?

Can be improved

Are the conclusions supported by the results?

Yes

Response and Revisions:

We sincerely appreciate the reviewer's insightful comments and suggestions. We appreciate that you found the introduction sufficiently informative, with relevant references included, and the research design appropriate. In the Materials and Methods section, we have elaborated on the characteristics of the viruses employed, added detailed parameters for the bioinformatics analysis, and provided a thorough description of the statistical analysis methods. We appreciate your affirmation that the conclusions are well supported by the data presented. The corresponding revisions, indicated in track changes, are included in the re-submitted files. We hope that the revisions will meet with your approval. We appreciate your guidance throughout this process.

3. Point-by-point response to Comments and Suggestions for Authors

Comments 1: Lines 122-126: The presentation of DEP numbers (732 DEPs: 298 increased, 434 decreased) should be accompanied by the statistical criteria used for classification in this initial mention.

Response 1:

We sincerely appreciate the reviewers’ insightful comments and suggestions. Fold change >1.50 or <0.67 and P<0.05 were set as the significance thresholds for screening of DEPs. The revised sentence in the manuscript now reads “Compared to the control group, the CK/C17-PB2/627K group showed 732 DEPs, comprising 298 proteins that were significantly increased (fold change > 1.50) and 434 proteins which were significantly decreased (fold change < 0.67), both at P < 0.05 (Figure 3 A).” in lines 140-142.

Comments 2: Lines 165-180: The functional enrichment analyses results need clearer organization. The "top 20 terms" are mentioned but the selection criteria for determining "top" status isn't explained.

Response 2:

We are deeply grateful for your insightful comments and constructive suggestions. In response to your feedback, we have provided a more detailed explanation of our enrichment analysis in Section 2.5, specifically in lines 183-188. The functional enrichments of differentially expressed proteins (DEPs) were evaluated using a two-tailed Fisher’s exact test, which is a robust statistical method for assessing overrepresentation. In the bubble diagrams, the vertical axis denotes the functional categories or pathways, while the horizontal axis represents the enrichment factor. The enrichment factor was measured with log2-transformed fold change of the proportion of DEPs in that functional type compared to the proportion of identified proteins. The "top 20 terms" were selected based on the highest log2 fold enrichment values, which reflect the most significant enrichments in our dataset. This approach ensures that the terms highlighted are those with the greatest impact on our analysis.

Comments 3: Lines 201-220: The rationale for selecting SNAPIN specifically for validation among the many DEPs identified isn't adequately explained. The authors should justify this choice.

Response 3:

We sincerely appreciate the reviewers’ insightful comments and suggestions. Autophagy plays a crucial role in the life cycle of IAVs and regulated by various host factors. In our preliminary literature review, we found that the SNAPIN protein could regulate autophagy, but whether this protein is involved in influenza virus-induced cellular autophagy and what its mechanism of action remains unknown. We are particularly interested in the role of SNAPIN in influenza virus-induced cellular autophagy and the viral replication cycle, so we have chosen to verify the expression changes of this protein after viral infection and plan to conduct in-depth studies on this protein in our subsequent work. The revised section was listed in lines 222-225.

Comments 4: Lines 335-343: The cell infection protocol description lacks important details:

- Specific viral titers used

- Standardization method for MOI across different virus strains

- Controls for viral infection efficiency

Response 4:

We sincerely appreciate the reviewers’ insightful comments and suggestions. The viral titers of CK/C17-PB2/627K, WSN, and CK/C88 were 1.7 × 108 PFU/mL, 2.4 × 108 PFU/mL, 2.6 × 107 PFU/mL, respectively. The MOI is calculated by dividing the number of viral particles by the number of host cells. In the context of proteomic analysis, A549 cells were grown in 10 cm dishes, with a cell count of 5.5 × 106 cells per dish at 90% confluence. To achieve an MOI of 1, 5.5 × 106 PFU of H9N2 virus were added to these dishes. For qPCR analysis of SNAPIN, cells were seeded in 6-well plates for infection with the specified viruses. At 90% confluence, the cell count was 9.8 × 105 cells per well, and to achieve an MOI of 1, 9.8 × 105 PFU of CK/C17-PB2/627K, WSN, and CK/C88 viruses were added to the 6-well plates. The revised sections were listed in lines 351-355. The reviewer has raised a very good question about the viral infection efficiency. We visualized viral NP protein using indirect immunofluorescence assay (IFA) to ensure the efficiency of viral infection before subjected to mass spectrometry analysis (Figure 1A). NP were detected in nearly all A549 cells infected with CK/C17-PB2/627K virus at an MOI of 1 at 24 hpi, while no NP was detected in the control group inoculated with pure chicken embryo allantoic fluid. The result indicated that the infection dose of virus in A549 cells was effective. We added this section to section 2.1 in lines 84-89.

Comments 5: Lines 366-383: The mass spectrometry methodology section could be improved by:

- Including details about sample preparation reproducibility

- Explaining the choice of MS parameters

- Describing quality control measures for MS data acquisition

Response 5:

We sincerely appreciate the reviewers’ insightful comments and suggestions. The virus treated or control samples for proteomic analysis were prepared with three replicates to ensure that the obtained data could be subjected to biological statistical analysis.

We have employed a rigorous and standardized sample preparation protocol to ensure the reproducibility of our results. This includes steps such as protein extraction, solubilization, reduction, alkylation, and tryptic digestion. Each step is carefully controlled to minimize variability and ensure consistency across samples. We used a lysis buffer containing a cocktail of protease and phosphatase inhibitors to prevent degradation and modification of proteins during the extraction process. The solubilization efficiency was monitored to ensure complete protein solubility, which is critical for downstream analysis. To enhance digestion efficiency, we employed Trypsin Protease of MS Grade. The digestion was performed under tightly controlled conditions to ensure complete and specific cleavage of proteins into peptides. We assessed the reproducibility of our sample preparation by performing multiple technical replicates. The consistency of the peptide profiles obtained from these replicates was analyzed using statistical methods to ensure the reliability of our sample preparation method. We implemented stringent quality control measures at each step of the sample preparation process. This includes regular calibration of equipment, use of certified reagents, and adherence to standardized protocols to minimize batch effects and ensure the reproducibility of our results.

We have carefully selected our MS parameters to optimize the sensitivity, specificity, and reproducibility of our analysis in lines 388-405. Below, we explain the rationale behind our choices. We chose a mass range of 400 to 1200 m/z, which is typical for peptide analysis. This range covers most peptide ions that result from tryptic digestion. We set a resolution of 60,000 in the primary mass spectrometry and 15,000 in the secondary mass spectrometry, these settings provide high mass accuracy and sensitivity. The signal threshold was set at 5×104 ions/s to ensure that only peptides with a significant signal intensity are selected for MS/MS analysis.

We have implemented a series of rigorous quality control measures to ensure the high quality and reliability of our MS data. We incorporate the use of quality control samples throughout our analysis to monitor and correct for any potential biases or variations in the data.

Comments 6: Lines 384-406: The protein quantification section needs:

- Clearer explanation of the normalization process

- More detail about how missing values were handled

- Better description of the statistical approach for determining significance

Response 6:

We sincerely appreciate the reviewers’ insightful comments and suggestions. In our study, we employed LFQ intensity to adjust variations across samples, ensuring the accuracy of protein quantification in lines 415-420. This process centralizes the intensity levels of proteins, allowing us to derive relative quantitative values (R) by correcting for potential biases introduced during sample processing, injection, and instrumental analysis. For label-free quantification, we employed the MaxQuant LFQ algorithm, which uses the area under the curve (extracted ion currents-XIC) for quantification and is efficient for large search spaces, including those with unspecific searches. This method inherently accounts for missing data by focusing on the presence or absence of peptides rather than their absolute quantity. We utilized the two-sample, two-tailed t-test to evaluate the statistical significance of the differential expression of proteins between experimental groups (lines 424-429). This test is appropriate for our data as it assumes unequal variances between the control and treatment for each protein, and this approach accounts for both increases and decreases in protein expression. Besides, the output of the t-test provides us with p-values for each protein, indicating the probability that any observed difference in protein expression between groups is due to chance. We set our significance threshold based on these p-values to determine which proteins show a statistically significant change in expression.

Comments 7: Lines 407-421: The bioinformatics analysis section requires:

- Version numbers for all software used

- Parameters used for network construction

- Cutoff values for significance in enrichment analyses

Response 7:

We sincerely appreciate the reviewers’ insightful comments and suggestions. The versions of all software used in this study were added in the manuscript in lines 432-435. Parameters used for PPI network construction were added to the text in lines 441-450. The GO and KEGG enrichment of the DEPs against all proteins of the species database was assessed using a two-tailed Fisher’s exact test. The pathway with a corrected P< 0.05 is considered statistically significant. We have added this information in section 4.7 in lines 437-440.

Comments 8: Lines 422-426: The mouse study description lacks:

- Power calculation for sample size determination

- Details about randomization

- Specifics about monitoring criteria

Response 8:

We sincerely appreciate the reviewers’ insightful comments and suggestions. Six mice were randomly divided into two groups evenly. Three mice were intranasally inoculated with 106 EID50 of the WSN virus in a volume of 50 μL, with three mice intranasally inoculated with PBS as a control. The lungs of the three mice were collected for detection of SNAPIN by using qPCR and Western blot 72 hpi. For analysis of qPCR, GAPDH was used as an endogenous control for mRNA normalization. Relative expression levels for each gene were calculated using the 2−∆∆Ct method. The revised section was in lines 453-456. The mice were monitored for 72 h, and only slight weight loss was observed in the infected group mice (data not shown).

Comments 9: Lines 427-437: The PCR methodology needs:

- Primer sequences

- PCR efficiency values

- Details about normalization controls

Response 9:

We sincerely appreciate the reviewers’ insightful comments and suggestions. RT-PCR and qPCR for detecting SNAPIN were performed with the following primers: forward primer 5’-AGGAACGACTGAGACGGCTAA-3’ and reverse primer 5’-GGTAAATTCCCGAATCCAGCATT-3’. RT-PCR for detecting NP was performed with the following primers: forward primer 5’-TCAGCATACAACCTACGTTC-3’ and reverse primer 5’-GCATTGTCTCCGAAGAAATAAG-3’. RT-PCR for detecting GAPDH was performed with the following primers: forward primer 5’- AGGTGAAGGTCGGAGTCAACG-3’ and reverse primer 5’-TGGAAGATGGTGATGGGATTTC -3’. These primers were added in section 4.9 in lines 460-474. A single distinct peak in the melting curve plot was detected in the qPCR analysis, indicating the high specificity of the amplification of the target DNA segment. A549 cells treated with PBS were subjected to negative controls for PCR and RT-PCR in figure 7.

Comments 10: Lines 438-446: The Western blot section should include:

- Information about loading controls

- Antibody validation details

- Quantification methodology

Response 10:

We sincerely appreciate the reviewers’ insightful comments and suggestions. Three mice were intranasally inoculated with WSN virus or PBS (as a nagetive control). Lungs were collected at 72 hpi for analysis of SNAPIN expression levels using Western blot. The lungs of the PBS inoculated mice were subjected to negative controls in Western blot analysis. β-actin was chosen as the loading control. The revised section was in line 241 and lines 476-480. Rabbit anti-SNAPIN polyclonal antibody (pAb) was purchased from Santa Cruz Bio-technology Inc. (sc-514675, Santa Cruz). The SNAPIN band detected in the membranes was between 19KD and 21KD according to the instructions (https://www.scbt.com/zh/p/snapin-antibody-e-10). The specific band between 15KD and 25KD was detected in our Western blot analysis. In addition, the rabbit anti-NP pAb was produced and stored in our laboratory. NP was detected slightly above 55KD in our Western blot analysis. These results demonstrate that the antibodies are effective and can be used to detect the corresponding proteins. For analysis of qPCR, GAPDH was used as an endogenous control for mRNA normalization. Relative expression levels for each gene were calculated using the 2−∆∆Ct method. The revised section was in lines 460-474.

Comments 11: Lines 450-459: The conclusions section could be strengthened by:

- Addressing study limitations

- Discussing potential therapeutic implications

- Suggesting specific future research directions

Response 11:

We sincerely appreciate the reviewers’ insightful comments and suggestions. We revised the conclusion section and added the limitation of this 501-516. The revised section now reads as follows: “However, it is important to acknowledge the limitations within our research scope. The interaction between IAVs and the host is a dynamic process. A proteomic analysis of samples collected 24 hpi was performed in this study; thus, these interaction networks may not fully represent the dynamic nature of the virus-host interaction. Despite identifying numerous DEPs using proteomic techniques, their involvement in the viral life cycle and specific roles require additional study. Further characterization of host proteins critical for viral replication could reveal potential targets for the development of broad-spectrum antiviral therapies. We propose that modulating the expression or activity of these DEPs could offer a new novel avenue for antiviral treatment. Specifically, the downregulation of SNAPIN upon IAV infection suggests a potential role in viral pathogenesis. Furthermore, investigating the cross-reactivity of these proteins with other viral strains could evaluate their potential for broad-spectrum antiviral therapy. Our study presents a detailed network for IAV infection in A549 cells, facilitating the screening of key host proteins which regulate viral replication, potentially offering functional targets for the development of broad-spectrum antiviral therapies.”

Comments 12: Lines 468-471: The funding statement should clarify the role of each funding source in the study design and execution.

Response 12: We sincerely appreciate the reviewers’ insightful comments and suggestions. We revised the funding statement in line 528 The funders had no roles in the study design and execution.”

Comments 13: Lines 94-102: Figure 1 requires:

- Higher resolution images

- Better explanation of the statistical analysis

- Clearer labeling of axes

Response 13:

We sincerely appreciate the reviewers’ insightful comments and suggestions. We have added clearer detailed labeling information to the axes of Figure 1. We have revised Figure 1 to make the image meet the standards of high resolution (600 dpi). Better explanations of the PCC and RSD analysis were revised in the manuscript in lines 97-108, which now reads as follows: “The Pearson's Correlation Coefficient (PCC) quantifies the linear correlation between the two data sets (control and CK/C17-PB2/627K virus infection group): scores approaching -1 denote a stronger negative correlation, scores approaching 1 denote a stronger positive correlation, and values near zero suggest a lack of correlation. PCC analysis demonstrated that the quantitative values nearing 1, indicating a positive correlation between the control and CK/C17-PB2/627K virus-infected group (Figure 1D). We plotted the boxplot based on the relative standard deviation (RSD) of protein quantification values among replicate samples. A lower overall RSD indicates superior quantitative consistency. The RSD distribution across replicates underscored the accuracy and reliability of our proteomic data, with the CK/C17-PB2/627K strain-exposed group exhibiting lower RSD values than the negative control group (Figure 1E).”

Comments 14: Lines 182-185: Figure 5 bubble diagrams need:

- Clearer size legends

- More detailed explanation of the enrichment scores

Response 14:

We sincerely appreciate the reviewers’ insightful comments and suggestions. We have added the detailed explanation of the enrichment parameters to the revised manuscript in lines 183-187. Now the revised section reads as following: “In the bubble diagrams, the vertical axis represents the functional categories or pathways, and the horizontal axis represents the log2-transformed fold change of the pro-portion of DEPs in that functional type compared to the proportion of identified proteins. The color of the circles indicates the enrichment significance P value, and the size of the circles represents the number of DEPs in the functional category or pathway.” We have revised Figure 5 to make the image meet the standards of high resolution (600 dpi).

Comments 15:

General formatting issues:

- Inconsistent abbreviation usage throughout the manuscript

- Some figure legends lack sufficient detail

- References need standardization in format  

Response 15:

We sincerely appreciate the reviewers’ insightful comments and suggestions. We have taken them seriously and made the following revisions to our manuscript. Firstly, we have meticulously revised the entire manuscript to ensure that abbreviations are used consistently. Each abbreviation is now clearly defined at its first occurrence within the text. Secondly, we have revised the legends for all figures to include more detailed information and context. This will enable readers to grasp the significance of the data presented without needing to refer to the main text, thus improving the self-sufficiency of the figures. We have also checked the figure qualities to make the image meet the standards of high resolution (600 dpi). Thirdly, we have standardized the format of all references to conform to the journal's guidelines. This includes adopting a uniform citation style and presentation, ensuring that all references are formatted consistently throughout the manuscript according to the requirements of IJMS. These revisions will significantly improve the clarity and professional presentation of our manuscript. We appreciate your guidance and are dedicated to addressing all aspects that enhance the quality of our submission.

4. Response to Comments on the Quality of English Language

Point 1: The quality of English does not limit my understanding of the research.

We are grateful for your assessment that the English quality of our manuscript is adequate for understanding. We try to ensure that our scientific contributions are communicated with utmost clarity. We have conducted a thorough review of the manuscript, making minor corrections to meet the rigorous standards of academic publishing and to improve readability for our audience. The changes have been highlighted with track changes in the revised manuscript.

Reviewer 2 Report

Comments and Suggestions for Authors

Dear authors

I hope you are all doing well. Regarding the revision of the manuscript No. ijms-3370944 entitled "Proteomic analysis of differentially expressed proteins in A549 cells infected with H9N2 avian influenza virus". It is really a very interesting and high-quality paper that aimed to study the differentially expressed proteins in A549 cells infected with H9N2 avian influenza virus. However, I have only 2 minor comments that should be answered.

Comments

1- Please specify the IAV subtype used to achieve the results of 2.1, 2.2, 2.3, 2.4 and 2.5. Which of the three viruses H9N2, H1N1 and H3N8? Please specify.

2- The in vivo protein expression in three mice was applied only to H1N1. Why weren't the H9N2 and H3N8 viruses included?

Author Response

For research article

Response to Reviewer 2 Comments

1. Summary

We are grateful to the reviewers for their meticulous review and insightful comments. We have found these comments to be invaluable, and the revisions implemented based on the reviewers' recommendations have markedly enhanced the quality of our manuscript. Below are our point-by-point responses to the reviewers' comments. The corresponding revisions, indicated in track changes, are included in the re-submitted manuscript. We hope that the manuscript now meets the criteria for publication in IJMS.

2. Questions for General Evaluation

Reviewer’s Evaluation

Does the introduction provide sufficient background and include all relevant references?

Yes

Is the research design appropriate?

Yes

Are the methods adequately described?

Yes

Are the results clearly presented?

Can be improved

Are the conclusions supported by the results?

Yes

Response and Revisions:

We sincerely appreciate the reviewer's insightful comments and suggestions. We are pleased that you found the introduction sufficiently informative, with relevant references included, the research design appropriate, and the methods adequately described. We also appreciate your affirmation that the conclusions are well supported by the data presented. We have revised the manuscript in accordance with the reviewers' comments. The corresponding revisions, indicated in track changes, are included in the re-submitted files. We hope that the revisions will meet with your approval. We remain committed to upholding the highest standards of research communication and appreciate your guidance throughout this process.

3. Point-by-point response to Comments and Suggestions for Authors

Dear authors, I hope you are all doing well. Regarding the revision of manuscript No. ijms-3370944 entitled "Proteomic analysis of differentially expressed proteins in A549 cells infected with H9N2 avian influenza virus". It is really a very interesting and high-quality paper that aimed to study the differentially expressed proteins in A549 cells infected with H9N2 avian influenza virus.

Comment 1: However, I have only 2 minor comments that should be answered. Please specify the IAV subtype used to achieve the results of 2.1, 2.2, 2.3, 2.4 and 2.5. Which of the three viruses H9N2, H1N1 and H3N8? Please specify.

Response 1:

We sincerely appreciate the reviewer’s insightful comments and suggestions. We concur with the suggestion that the specific names of the IAVs should be clearly presented in the manuscript. Detailed information about the H9N2, H1N1, and H3N8 viruses has been presented in section 4.2 in lines 351-355. To preclude any confusions, we have adopted these specific virus designations throughout the manuscript and figures.

Comments 2: The in vivo protein expression in three mice was applied only to H1N1. Why weren't the H9N2 and H3N8 viruses included?

Response 2:

We sincerely appreciate the reviewer’s insightful comments and suggestions. The reviewer has raised a very pertinent question. The host protein SNAPIN was found to be downregulated following infection with CK/C17-PB2/627K virus in our proteomic analysis. The downregulation of SNAPIN was confirmed via qPCR analysis of the mRNA levels of SNAPIN post-infection with CK/C17-PB2/627K, WSN, and CK/C88 viruses. These results suggest that the regulation of SNAPIN may be a universal phenomenon after infection with IAVs in vitro. Thus, we selected the laboratory model influenza virus strain WSN to perform the mice experiment. As anticipated, SNAPIN expression was also downregulated 72 hpi following infection with WSN virus in mice. It is essential to assess the response of SNAPIN to IAV infection in vivo to determine if this expression pattern is a universal phenomenon across different subtype viruses, including H9N2 and H3N8. We will address this content in our forthcoming studies on the function and mechanism of SNAPIN in regulating IAV replication.

4. Response to Comments on the Quality of English Language

Point 1: The quality of English does not limit my understanding of the research.

Response:

We sincerely appreciate the reviewer’s indication that the English quality in our manuscript is sufficient for comprehension. We strive to ensure that our scientific contributions are communicated with utmost clarity. We have meticulously reviewed the manuscript and corrected minor errors to ensure it adheres to the high standards of academic publishing and to enhance comprehension for all readers. The revised sections are indicated using track changes in the manuscript.

Reviewer 3 Report

Comments and Suggestions for Authors

The article, Proteomic characterisation of differentially expressed proteins in A549 cells infected with H9N2 avian influenza virus, provides a thorough proteomic assessment of host cellular protein changes following H9N2 infection. However, numerous important concerns might be raised:

1.      The study's findings are substantially consistent with prior publications. Notably:

      • https://pmc.ncbi.nlm.nih.gov/articles/PMC5156691/
      • https://virologyj.biomedcentral.com/articles/10.1186/s12985-021-01512-4

2.      The study focusses on SNAPIN as a novel regulator of IAV replication, however its involvement in autophagy and viral pathogenesis has been highlighted in prior research, such as PRRSV and HCMV.

3.      Despite its robustness, the 4D label-free proteomics methodology does not offer unique insights or methodologies for analysing IAV-infected cells.

4.      GO and KEGG analyses are typical and consistent with previous research findings. Cytokine signalling, protein breakdown, and autophagy are all well-documented mechanisms in IAV.

5.      The paper supports SNAPIN expression but does not include mechanistic or functional research to understand its involvement in IAV replication. The paper depends mainly on statistical correlations, with no experimental validation beyond basic proteomics.

Comments on the Quality of English Language

The English could be improved to more clearly express the research.

Author Response

For research article

Response to Reviewer 3 Comments

1. Summary

We thank the reviewers for a very careful review and thoughtful comments. These comments are very helpful, and the revisions made as per the reviewers’ suggestions have significantly improved the quality of our manuscript. Below are point-by-point responses to the reviewers’ comments. The corresponding revisions in track changes are in the re-submitted files. We hope that the manuscript is now suitable for publication in IJMS.

2. Questions for General Evaluation

Reviewer’s Evaluation

Does the introduction provide sufficient background and include all relevant references?

Must be improved

Is the research design appropriate?

Must be improved

Are the methods adequately described?

Must be improved

Are the results clearly presented?

Must be improved

Are the conclusions supported by the results?

Must be improved

Response and Revisions:

We sincerely appreciate the reviewer’s comments and suggestions for the thorough evaluation to improve our manuscript. We have expanded the introduction to provide more comprehensive background information and have added relevant references to ensure that our literature review is up-to-date and encompasses the most pertinent studies in the field in lines 37-39 and lines 72-75. We have revised our research design to ensure it is more rigorous and appropriate for the questions we are addressing. This includes clarifying the rationale behind our experimental approach and making any necessary adjustments to enhance the validity and reliability of our study. In response to your feedback, we have detailed our methods section 4 to provide a clearer and more step-by-step description of the procedures followed. This includes additional information on the virus and agents used, the specific protocols adhered to, and statistical analysis in the revised manuscript. Figures and textual descriptions have been refined to ensure that the data are presented in a logical and easily interpretable manner, facilitating the reader's understanding of our findings. To strengthen the conclusions drawn from our results, we have re-organized the language and ensured that every conclusion is directly supported by the results presented. We have also made sure to discuss any limitations and the implications of our findings within the context of the broader research landscape. We believe that these revisions have significantly improved the quality and clarity of our manuscript.

3. Point-by-point response to Comments and Suggestions for Authors

Comments 1: The article, Proteomic characterization of differentially expressed proteins in A549 cells infected with H9N2 avian influenza virus, provides a thorough proteomic assessment of host cellular protein changes following H9N2 infection. However, numerous important concerns might be raised:

The study's findings are substantially consistent with prior publications. Notably:

    https://pmc.ncbi.nlm.nih.gov/articles/PMC5156691IF: 4.0 Q2 /

https://virologyj.biomedcentral.com/articles/10.1186/s12985-021-01512-4IF: 4.0 Q2

Response 1:

We sincerely appreciate the reviewer's valuable comments, which provide constructive feedback for enhancing the quality of our manuscript. We concur with the reviewer's observations. It is indeed accurate that prior studies have examined the differential expression of host proteins in A549 cells infected with the H9N2 virus. Upon careful review of the two articles provided by the reviewer, we identified that our study differs from these two in several respects:

First, the experimental methodologies employed differ significantly. Both published studies utilized two-dimensional electrophoresis (2-DE) for protein sample separation, followed by mass spectrometry for the identification of differentially expressed proteins. Our study employed 4-D label-free technology for the analysis of whole cell lysates. While both techniques are widely applied, the 2-DE method is limited by its low throughput, low reproducibility, and inability to detect proteins with high and low abundance. Additionally, it exhibits poor performance in analyzing hydrophobic proteins, membrane proteins, and proteins with extreme isoelectric point (pI) values. In contrast, the 4-D label-free technology utilized in this study offers a broader detection range, enhanced sensitivity and throughput, and an expanded dynamic range, yielding precise quantitative data. Yu and colleagues identified only 16 DEPs before and after virus infection; Yang and colleagues identified 32 differentially expressed proteins after isolating cell mitochondria; whereas our study, employing 4-D label-free technology, identified 732 differentially expressed proteins, providing a more comprehensive reflection of the impact of virus infection in cells.

Second, the model virus strains differ: The H9N2 and H5N1 virus strains in the published studies are avian influenza viruses, whereas our study employs a strain adapted to A549 cells, featuring the key amino acid mutation PB2-E627K in the PB2 protein for mammalian adaptation. Variations in infecting strains may result in distinct patterns of differential protein expression within the cells. The DEPs identified in our study may facilitate the screening of potential key proteins that affect influenza virus replication. We believe these distinctions underscore the unique contributions of our study and enhance our understanding of the host response to H9N2 virus infection. We cited these two papers to illustrate the limitations of their studies in our manuscript in lines 72-75.

Comments 2: The study focusses on SNAPIN as a novel regulator of IAV replication, however its involvement in autophagy and viral pathogenesis has been highlighted in prior research, such as PRRSV and HCMV.

Response 2:

We sincerely appreciate the reviewer's insightful comments. Thank you for your astute observation regarding the previously reported roles of SNAPIN in other viral infections. We agree that the involvement of SNAPIN in autophagy and viral pathogenesis, as observed with PRRSV and HCMV, highlights the importance of our identification of this protein as a differentially expressed candidate in IAV infection. Although autophagy plays a crucial role in the influenza virus life cycle and several host proteins are involved, unresolved questions remain. The involvement of SNAPIN in autophagy and its potential roles in IAV, previously unexplored, warrants further investigations. Furthermore, in addition to SNAPIN protein, other DEP candidates are also involved in our future studies.

Comments 3: Despite its robustness, the 4D label-free proteomics methodology does not offer unique insights or methodologies for analyzing IAV-infected cells.

Response 3:

We sincerely appreciate reviewer's feedback on the application of the 4D label-free proteomics methodology in our study. While the methodology itself may not be entirely novel, the depth and breadth of our proteome coverage provides a comprehensive view of the cellular changes induced by IAV infection. We understand that the field of proteomics is rapidly evolving, and it is crucial to demonstrate the distinctive contributions of our study. Compared to methods like two-dimensional electrophoresis (2-DE) coupled with mass spectrometry, we have identified a significantly larger number of DEPs using the 4D label-free proteomics, enriching our dataset for further in-depth analysis. Bioinformatic analysis enables us to discern complex patterns and trends in protein expression that might be obscured in less sophisticated analyses. By identifying key proteins and pathways that are modulated during IAV infection, we provide a foundation for the rational investigations of IAVs. To further address your concerns, we have revised the manuscript to more clearly highlight these points and to underscore the significance of our findings within the broader context of IAV research. We have also included a more detailed discussion on the limitations and the potential for future studies to build upon our work. We believe that the cumulative knowledge gained from such studies is invaluable for the scientific community. We hope that these revisions address your concerns and demonstrate the value of our study.

Comments 4: GO and KEGG analyses are typical and consistent with previous research findings. Cytokine signaling, protein breakdown, and autophagy are all well-documented mechanisms in IAV.

Response 4:

We sincerely appreciate the reviewer's insightful comments. We concur that cytokine signaling, protein breakdown, and autophagy are indeed well-documented mechanisms in the context of IAV infection. It is reassuring to know that our results align with the established body of scientific knowledge. While we acknowledge that the novelty of our findings may not be groundbreaking, the alignment with existing literature is instrumental in substantiating the reliability of our proteomics data. This concordance not only reinforces the robustness of our methodologies but also bolsters the credibility of our results. Moving forward, we plan to focus on less-studied differentially expressed proteins to explore new avenues of research. Furthermore, in addition to SNAPIN protein, other crucial less-studied DEP candidates are also involved in our future studies.

Comments 5: The paper supports SNAPIN expression but does not include mechanistic or functional research to understand its involvement in IAV replication. The paper depends mainly on statistical correlations, with no experimental validation beyond basic proteomics.

Response 5:

We sincerely appreciate the reviewer's insightful comments. We acknowledge your points regarding the need for mechanistic and functional research to better understand the involvement of SNAPIN in IAV replication, as well as the reliance on statistical correlations in our current study. In this study, our strategy was to map out a broad proteomic profile, thereby setting a solid foundation for future investigative work. The inclusion of SNAPIN was indeed to affirm the accuracy of our proteomics findings. We are planning to expand our research to include mechanistic studies that will provide a deeper understanding of SNAPIN's role in IAV replication. This will involve experimental validations that go beyond basic proteomics to offer a more comprehensive view of the molecular interactions and functions of SNAPIN.

4. Response to Comments on the Quality of English Language

Point 1: The English could be improved to more clearly express the research.

Response 1:

Thank you for your suggestion to enhance the clarity of the English in our manuscript. We are committed to ensuring that our research is presented with the greatest possible clarity and precision. To address this feedback, we have carefully reviewed the entire manuscript and made targeted improvements to the language and expression. Our aim was to refine the text to meet the stringent standards of academic publishing and to ensure that the content is accessible and understandable to our readership. The changes have been implemented using track changes, allowing for a clear visualization of the revisions made. We trust that these amendments will significantly enhance the manuscript's readability and the precision of the scientific discourse. Your guidance is invaluable in helping us achieve the highest quality in our submission.
